# To Explain or Not to Explain: A Study on the Necessity of Explanations for Autonomous Vehicles

**Yuan Shen**[1]   **Shanduojiao Jiang**[2]   **Yanlin Chen**[3]   **Katie Driggs-Campbell**[1]
[1]University of Illinois at Urbana-Champaign   [2]Stanford University   [3]Apple Inc.
{yshen47, krdc}@illinois.edu
sj99@stanford.edu   yanlinchen@outlook.com

## Abstract

Explainable AI, in the context of autonomous systems, like self-driving cars, has drawn broad interests from researchers. Recent studies have found that providing explanations for autonomous vehicles' actions has many benefits (e.g., increased calibrated trust and acceptance), but put little emphasis on when an explanation is needed and how the content of explanation changes with driving context. In this work, we investigate which scenarios people need explanations and how the critical degree of explanation shifts with situations and driver types. Through a user experiment, we ask participants to evaluate how necessary an explanation is and measure the impact on their trust in self-driving cars in different contexts. Moreover, we present a self-driving explanation dataset with first-person explanations and associated measures of the necessity for 1103 video clips, augmenting the Berkeley Deep Drive Attention dataset. Our research reveals that driver types and driving scenarios dictate whether an explanation is necessary. In particular, people tend to agree on the necessity for near-crash events but hold different opinions on ordinary or anomalous driving situations.

## 1 Introduction

Artificial intelligence (AI) is becoming prevalent in everyday life, powering smart devices, personalizing assistants, and enabling autonomous vehicles. However, people encounter difficulties to accept, understand, or trust this technology [1]. This phenomenon is especially true in the self-driving car industry, where people are hesitant to hand over control of the steering wheel to AI [2, 3]. One of the reasons for this distrust is that people are uncertain about how those sophisticated models make car control decisions [4]. With the ambiguity in the decision process, it is hard for passengers to tell whether the model makes the right judgment given the current situation. The consequence of this ambiguity includes financial loss, legal issues, and even loss of human lives [5].

To improve the understanding of the self-driving car decision process, researchers have explored different perspectives to investigate explainable AI (XAI) in the autonomous vehicle domain [6, 7]. For example, the state-of-the-art research has shown that introducing explanations for self-driving control decisions can increase trust [8]. The study found that people trust (and prefer) explanations presented before the car takes an action, compared to after-action, no explanation, and intervention-based explanations. Others argued that explanations for autonomous vehicles should be modified based on context [9].

To generate text-based descriptions, researchers collected the Berkeley DeepDrive Explanation Dataset, where they extensively annotated every single car decision with explanations [10]. However, the existing research over-emphasized the benefits of introducing explanations for self-driving cars and neglected the possibility that people might not need those explanations during certain scenarios.

Progress and Challenges in Building Trustworthy Embodied AI workshop paper at 36th Conference on Neural Information Processing Systems (NeurIPS 2022).

In other words, it is still unclear *when* it is necessary to introduce explanations about the autonomous decision, and whether we should address individuals differently.

In this work, we examine when and how explanations should be presented to users of autonomous vehicles. Specifically, we investigate which scenarios people need explanations for and how the critical degree of an explanation shifts with situations and driver types. We focus on text-based descriptions with varying content to assess what information and narrative is preferred. Our findings are validated in a survey-based user experiment online, in which subjects imagine themselves as passengers of the vehicles in driving video clips of a variety of scenarios. For each video, we record each participants' reported explanation necessity rating, attentiveness score, and preferred explanation content. During the post-survey, we collect participants' responses on their driver types and general trust level on autonomous vehicles.

Using the data collected, we aim to understand the relationship between driver types and the necessity of an explanation, for particular contexts. Specifically, in early tests, we observed disagreements among participants on the explanation necessity level. For instance, we found that people tend to agree on that explanations are necessary in near-crash events, but there was no obvious agreement for ordinary or anomalous driving situations in aggregate. As we will show, when examining factors like driver type (i.e., cautious, aggressive) and context, a relationship is uncovered linking the necessity of explanations with the scenario and driver type.

To explore the explanation necessity level for more diverse scenarios, we present a self-driving explanation dataset by augmenting the Berkeley Deep Drive Attention dataset [11]. Each video in the dataset is annotated with explanation content, the explanation time interval, and associated measures of the necessity for all 1103 video clips. The associated explanation necessity score ranges from 0 to 1, suggesting how critical an explanation is needed for the given scenarios.

In summary, our contributions are as follows: (1) We present a study on the explanation necessity for autonomous vehicles; and (2) We present a dataset with video clips labeled with an explanation necessity degree, an explanation moment, and first-person explanation content.

## 2 Related work

### 2.1 Timing of Explanations

Apart from the forms and contents of explanations, we argue that it is also crucial to explore the timing of explanations. Timing for warnings and potential interventions is a key concern for (semi-) autonomous vehicles [12], motivating the fact the identification of key events is crucial for trustworthy autonomy. Koo et al. claim that it is critical to provide information to drivers/passengers ahead of an event [13]. Haspiel et al. designed a user experiment that introduces the importance of timing explanations in promoting trust in AVs [14]. Their study has discovered a pattern that suggests that explanations provided before the AV action promote more trust than explanations provided after.

Although existing studies have come to the agreement that explanations are more meaningful when put before an autonomous vehicle action, they failed to take into account that the necessities of providing explanations vary in different driving scenarios. In our experiment, we introduced a "critical score," which is a number ranging between 0 and 1, indicating how necessary an explanation is needed at each timestamp.

### 2.2 Definition of Critical Score

In previous studies about the relationship between human and autonomous vehicles, there have been various definitions of a critical score, namely, how critical a driving situation is for the passenger. Notably, Yurtsever et al. asked ten participants to watch driving video clips and give a score (subjectively) for the risk of maneuver seen in the videos [15]. After normalizing each annotators' ratings and taking the mean score as the final risk rating, they defined the top 5% of the riskiest videos as risky. However, it is necessary to distinguish "critical" (subjective) driving scenarios from "accident likely" (objective) situations. In another study, it has been proven that the perceived risk by humans is not necessarily proportional to the actual collision or accident probability associated with a specific driving situation [16]. Keeping these in mind, we proposed a human-centric, XAI-friendly definition of critical score: the necessity of explanation related to a particular driving maneuver.

### 2.3 Self-driving Explanation Dataset

The existing explanation dataset for self-driving suffers from various issues. For example, Berkeley DeepDrive eXplanation Dataset exhaustively labeled 6970 driving clips with explanations in specified video intervals [10]. However, a large portion of their driving clips is uneventful samples (e.g., cruising on the highway with constant speed), where humans require little need for the self-driving system to explain the situations. Furthermore, portions of some video clips are anomalous where the drivers do not follow the traffic rules (e.g., does not stop at a stop sign), thus have a poor (or illogical) explanation given the rules of the road. Meanwhile, as the explanations focused on *describing* the car model's decisions, the explanation content may not be ideal to promote a smooth conversation.

## 3 User Study

To understand people's need for a self-driving explanation for different scenarios, we conducted an online survey-based experiment. The experiment took 40 minutes, where we showed participants driving video clips and collected their responses. The goal of this experiment is to understand: (1) how necessary is a text-based explanation about self-driving car actions for different scenarios; (2) what the generally preferred explanation content is and if this is related to context; and (3) the relationship between user trust and explanations for autonomous vehicles.

### 3.1 Hypothesis

In our experiment, we target the following outcomes (dependent variables): explanation necessity, preferred explanation content, and user trust. We manipulate, vary, or estimate the following independent variables and influential factors: driving scenarios; explanation content (cause, effect, narrative type, Table 1); and presence of explanations. Based on the above three sets of dependent / independent variables, we derived three hypotheses:

1. Explanation necessity is correlated with driver types, and driving scenarios.
2. User's preferred explanation content is dependent on driving scenarios.
3. The presence of explanations will increase a user's trust in the automated vehicle.

### 3.2 Participants

In total, we have 18 participants for this user experiment. The majority of our participants are US college students with ages between 18 and 25. Among the participants, their driving experience is evenly distributed from 0.5 to 6 years.

### 3.3 Sampling Strategy

The driving video clips are sampled from our explanation dataset, described in the Dataset section. To capture a variety of typical scenarios, we conducted a text-based clustering using our annotated explanations for each video clip in our dataset. Our goal is to capture the different but representative scenarios in the dataset. We used hierarchical clustering with average linkage [17]. Specifically, we pre-processed the input text by lowercase and converting to TF-IDF score [18]. We used cosine similarity for the distance metric of the clustering. In the end, we used the videos in the 38 cluster centers for the user experiment. Our scenarios cover merging, parking, speed-up, near-crash, traffic-light, and stop-sign events.

### 3.4 Study Design & Procedure

Our user study is conducted as an online survey-based study. The experiment took 40 minutes to finish, including a break every fifteen minutes.

During the experiment, our participants watched 38 independent short driving video clips (as described in the previous subsection). Participants were told to imagine themselves as passengers riding in the vehicles to simulate the in-car experience. The video shown to participants was the raw video, without an explanation. Moreover, we randomized the order of sampled videos to reduce bias, and each

Table 1: Formats of explanation contents.

| Format | Narrative | Example |
|---|---|---|
| action + reason | first-person | I'll slow down for a broken traffic light. |
| action + reason | third-person | The car will slow down for a pedestrian. |
| action | first-person | I'll slow down. |
| action | third-person | The car is about to slow down. |

participant watched all sampled videos at the end of the experiment. After each video, participants answered several follow-up questions. The users rated how necessary an explanation is for the clip, referred to as a necessity score. Then, they described how attentive they were while watching the clip. Finally, they ranked several explanation candidates that we had prepared for each of the sampled videos. In particular, we prepared four different types of explanation contents separately for all of the 38 driving scenarios, as presented in Table 1. During the post-study phase, we prepared several questions related to how the text-based explanation can affect people's trust in autonomous vehicles, and to what the participant's driver types were.

## 3.5 Quantitative Analysis

We summarized our findings in three different aspects:

1. A correlation analysis between different scenarios and the reported explanation necessity level using Pearson correlation and Point-Biserial Correlation was performed [19, 20].
2. We performed a Friedman test to check the distribution of different explanation options [21] to test whether there is a global preferred explanation content format.
3. We analyzed the relationship between the presence of explanations and trust

To identify driver types, we consider participants aggressive if they satisfy any of the following condition:(1) The actual driving speed usually is above 35 mph for the road with the speed limit at 30 mph (2) They describe their driving type as aggressive explicitly in the post-study session (3) They report changing lane frequently even if unnecessary. Otherwise, we consider participants as cautious.

### 3.5.1 General Statistics

From our post-study questions, we learned some general views from our participants on autonomous vehicles. Our participants expressed general doubts on the feasibility of autonomous vehicles, where 77.8% of participants expressed a low level of trust in autonomous vehicles. Among our participants, only 10.6% of people believe the self-driving techniques will be readily available to the public in the next two years. Despite the distrust on the technology, participants expressed overall trust in the reliability of explanations from autonomous vehicles - 72.2% of the participants rated scores higher than five from a 1 to 10 Likert scale. This finding suggests introducing the right explanation contents has the potential to influence people's trust in autonomous vehicles.

From the driving clip questions, we learned that the average of explanation necessity level for each of the 38 driving scenarios ranges from 2.27 to 8.22, in a 1 to 10 Likert scale. We observed that people generally disagree on how necessary an explanation is needed for the same scenario, with an average standard deviation at 2.97 across 38 driving scenarios. Furthermore, we observed that the average explanation necessity score given by aggressive drivers is 18% lower than the cautious driver.

### 3.5.2 Correlation Analysis on Explanation Necessity

We did correlation analysis on the explanation necessity level on driver types and driving scenarios.

For driver types, in terms of aggressive or conservative, we used the Point-Biserial Correlation to calculate the correlation between explanation necessity and driver type [20]. The resulting correlation is -0.14, which means that there exists a negative relationship between being an aggressive driver and the need for scenario explanations. In other words, the more aggressive a driver is, the less he or she needs an account from the vehicle.

Table 2: Correlation between necessity and scenarios measured in Point-Biserial Correlation score.

| Scenarios | Correlation with Necessity |
|---|---|
| Near Crash | 0.1818 |
| Merge/Switch Lane | 0.1218 |
| Slow down | 0.1134 |
| Parking | 0.0979 |
| Pedestrians | 0.0235 |
| Stop Sign | 0.0145 |

Finally, for driving scenarios, we tagged binary attributes for the 38 driving scenarios in advance (e.g., whether the video is a near-crash scenario). Then, we used Point-Biserial Correlation to calculate the correlation between explanation necessity and the corresponding scenario in Table 2 [20]. For example, the first row shows that an explanation is highly necessary for near-crash situations.

The result of the correlations of explanation necessity with driver types and driving scenarios proves that our hypothesis 1 - explanation necessity is correlated with driver types and scenarios - holds.

### 3.5.3 Explanation Content Preference

To investigate if there is a generally preferred explanation (Table 1) format across all scenarios, we performed Friedman tests separately for different driving situations to deal with the ranking data of explanation contents [21]. Our null hypothesis, $h_o$ is that there is no difference for different explanation contents. We set the $\alpha$ level to be 0.05. According to our result, only 16 out of 38 scenarios reject the null hypothesis. However, we did not find those 16 scenarios sharing quantitative attributes based on our data. Therefore, we concluded that there is no globally preferred explanation format for self-driving scenarios, and thus our hypothesis 2 (Explanation content is correlated with driving scenarios) does not hold.

### 3.6 Qualitative Analysis

Besides the quantitative analysis above, we performed a qualitative study on the relationship between scenarios and explanation necessity. We noticed that the standard deviation for explanation necessity scores given by the participants varies a lot for different situations. To investigate whether there are common properties among scenarios that people mostly agree or disagree on, we compared the scene that has the highest standard deviation of explanation necessity with the situation that has the lowest standard deviation of explanation necessity, as shown in Figure 1. From this comparison, we observed that people tend to agree on the near-crash/emergent driving situations (e.g., cars cutting-in suddenly). On the other hand, for the ordinary scenarios (e.g., driving smoothly on the highway), people's opinion on explanation necessity varies.

## 4 Dataset

We present an explanation necessity dataset for autonomous vehicles. We limit our scope of explanation format to text-based explanation. The purpose of this dataset is to provide precise, case-by-case, and first-person perspective explanations that resolve the following issues: (1) explantion timing, i.e., when people need a reason for a driving decision; (2) explanation necessity, i.e., how critical the explanation should be; and (3) explanation content, i.e. what explanation people expect to be notified.

### 4.1 Data Source

We sample driving video clips from the Berkeley DeepDrive Attention (BDD-A) dataset [11], which contains 1232 braking event driving videos captured by a front-mirror dashcam. BDD-A provides diverse data that can potentially be used to model explanation necessity, including human gaze, car speed, and GPS metadata per frame. Even though the dataset size of the BDD-X dataset is six times greater than the BDD-A dataset, through our empirical analysis, we find a large portion of the videos are ordinary driving scenes (e.g., driving on the highway), which do not contain moments worth

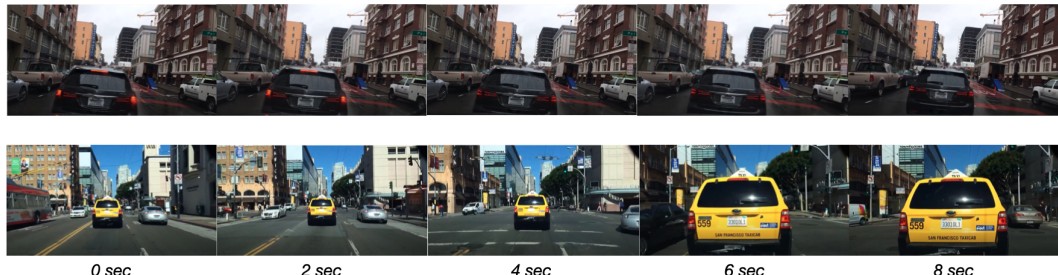

| 0 sec | 2 sec | 4 sec | 6 sec | 8 sec |

Figure 1: Frame sequences for videos with the highest and lowest standard deviation on the necessity for explanation. The first row corresponds to the video with the highest std, 3.41, and the second row corresponds to the one with the lowest std, 2.11. The average standard deviation for the 38 video clips is 2.97. We noticed that people are easier to reach an agreement on the explanation necessity for the near-crash/emergent driving situations. For example, participants gave similar explanation necessity for the second row, which describes a sudden brake between 4 seconds and 6 seconds. However, for those ordinary or not emergent driving situations, people's opinions for explanations vary a lot. For instance, participants have different views on the frame sequence in the first row, which is about a car gradually speeding up.

explaining explicitly. And the average video duration of the BDD-X is around 30 to 60 seconds, which is much longer for the BDD-A dataset, whose video usually lasts approximately 10 seconds.

## 4.2  Dataset Assumption

Our high-level assumption is similar to the data collection assumption proposed by Xia Y., et al. [11], where participants imagine themselves in the car of the driving videos. Specifically, we made the following assumptions:

1. The actual driver in the video follows the traffic laws.
2. The car action in the video is safe. In other words, the car action should not put the passengers into a high-risk car accident.
3. The recipient of the explanation would be a passenger of a fully autonomous vehicle. This assumption means that the perspective and sense/ability of control is different.
4. Every driving clips has at least one explanation moment. But the degree of explanation necessity can vary.

If any video clip fails to obey any of the assumptions, we removed it from our dataset.

## 4.3  Dataset Statistics

Our dataset contains 1103 driving video clips in total. From the video clips in the BDD-A dataset, we filtered out driving clips that did not meet our assumption criteria (e.g., drivers did not follow the traffic rules, poor videos quality like skipping frames) [11]. Five annotators were recruited for this dataset; the background of the annotators is college students age between 18 to 25 years old who have a valid driving license in the US.

## 4.4  Data Collection Procedure

For each of the video clips, the data annotators started by playing the driving video clips on an iOS mobile app, as shown in Figure 2. Once they reached a point where they considered the driving scene needed an explanation, the data annotators clicked on the *Record* button. Then, a pop-up window was presented to ask the participants to give a floating score, indicating how necessary people need an explanation for this moment based on their judgment. Additionally, they were asked to enter an text explanation at the current situations. Finally, annotators fine-tuned the timestamp to reflect the moment that needs an explanation. The cases were removed in which either the driver does

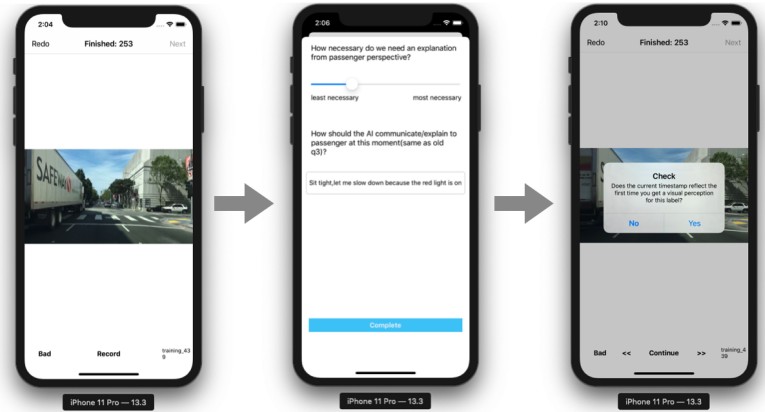

Figure 2: Workflow demo on our annotation iOS app. For each of the video clips, the data annotators clicked on *Record* button at the moment that needs an explanation from the self-driving system. Then, they entered a score indicating how critical an explanation is needed and gave a human-centered explanation. Finally, when they were directed back to the video page, they double-checked and fine-tuned the explanation moments.

Table 3: Dataset Content.

| Attribute | Type | Content |
| --- | --- | --- |
| vid | string | video id |
| message | string | text explanation |
| gazemap | mp4 | human gazemap at each timestamp |
| video | mp4 | driving video |
| necessity score | float | necessity degree for explanation |
| speed | float | car speed at each timestamp |
| course | float | car speed at each timestamp |
| explanation interval | float tuple | time segments for explanation to occur |

not follow traffic rules, or the driver action is too risky to be considered as the desired self-driving behaviors. For the explanation moment, we focused on recording before-action explanation, because previous research indicated people trust more on the before-action interpretation in autonomous vehicle settings [14].

### 4.5 Data Post-Processing Steps

To extract a general explanation necessity score from different people's reactions per example, we used truncated mean, a statistical measure of central tendency. In specific, for each data piece, we calculated the average of the explanation necessity scores after discarding the highest and lowest score. The advantage of the truncated mean is that it can reduce the influence of extreme scores. As for the explanation timestamp for each video example, we sorted the records annotated by different people for the corresponding explanation event. In other words, the format of the explanation time is a time interval that captures the moments where the relevant explanations should occur.

### 4.6 Data Model

Every video clips is annotated with one explanation moment. Each explanation contains a time interval that the explanation should take place, a first-person perspective explanation, and an explanation necessity score, indicating how critical an explanation is needed at one moment (Table 3). Instead of a binary response, the explanation necessity score is a floating number ranging from 0 to 1. To get a generalized explanation score for each driving clips, we collected responses from 5 different people and used truncated mean to get a general *critical score*.

Table 4: Model Performance in test set AUC score. $p_0$ stands for explanation necessity threshold.

| $p_0$ | random guess | our model |
|-----|--------------|-----------|
| 0.5 | 0.5803 | 0.6295 |
| 0.6 | 0.5427 | 0.6438 |
| 0.7 | 0.5359 | 0.6794 |

## 4.7 Explanation Necessity Inference

To demonstrate the potential of our dataset, we designed a spatial-temporal recurrent neural network that consumes a sliding window of RGB frames and classifies whether or not passengers need an explanation at the end timestamp of the input window. We prepare the ground-truth labels by manually defining the explanation necessity score threshold on annotated average necessity score. We evaluate our baseline model performance on test set AUC, and compare it against random guessing. The result in Table 4 demonstrates the feasibility of explanation necessity inference in real-time, even though a larger amount of dataset and a higher capacity model are necessary to obtain better performance.

## 5 Discussion

This paper investigates in-depth about when explanations are necessary for fully automated vehicles. There are two main aspects of results that are interesting for discussion.

Our initial hypothesis is that people would reach a certain amount of agreement on the explanation necessity level for different scenarios. However, our user study results indicate that people's opinions on explanation necessity might not share. Through our qualitative study, people tend to agree on the explanation necessity more for near-crash/emergent driving scenarios and less for the ordinary driving situation. Through our quantitative research, we found that both contexts and individual attributes had a significant influence on the desire for explanation.

Second, we previously considered that explanation of necessity should be highly correlated with speed changes. In other words, passengers should be more likely to ask for an explanation during speed decreasing moments. However, our correlation analysis between explanation necessity level and scenario types shows that this is not necessarily true. From Table 2, we noticed scenarios related to stop signs have the lowest correlation with necessity, even though the car is expected to decrease speed whenever approaching a stop sign. However, if we look at the scenarios that have a higher correlation with explanation necessity, they are, more or less, events that are not aligned with the passenger's original expectation. In other words, how different the scenario is from the expectation of passengers might be positively related to the explanation necessity level.

Our work is not without limitations. We quantified the explanation necessity level for different scenarios by collecting people's ratings directly. However, due to difference in individual criteria for explanation necessity, the recorded explanation necessity for scenarios have relatively high standard deviations that makes it hard to argue the general explanation necessity level for a given situation. This negative result gives insight into open problems in personalized explanation models for effective deployment. Moreover, a linear representation to capture explanation necessity is problematic, since the necessity level can be in a high-dimension space where each dimension corresponds to a different factor to necessity.

## 6 Conclusion

In this paper, we deeply investigated the necessity of explanations for autonomous vehicles. Our user experiment results showed that the need for explanation depends on specific driving scenarios and passenger identities. Along with this paper, we presented a self-driving explanation necessity dataset with first-person explanations and associated measure of necessity for 1103 video clips, augmenting the Berkeley Deep Drive Attention dataset. We hope that this dataset will help promote new research directions in explainability and intelligent vehicle design.

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
