# OpenReview forum: "To Explain or Not to Explain: A Study on the Necessity of Explanations for Autonomous Vehicles"
_NeurIPS.cc/2022/Workshop/TEA — TEA_

### Official Review · Reviewer_r532 · 2022-10-16
**A Paper on a Study of Explanation Necessity for Autonomous Vehicles**

**Rating:** 6
**Confidence:** 3

**Review:**

Summary:
In this paper, the authors performed a user study to understand people's need for explanations under various self-driving scenarios. They found that explanation necessity depends on attentiveness, driver types, and driving scenarios. Based on their findings, they augmented the Berkeley DeepDrive Attention (BDD-A) dataset by collecting study participants' explanation necessity scores, timestamps needed for explanation, and text explanations.

Strengths:
- The paper is clearly written and easy to follow.
- The paper is well-motivated.
- The paper has made contributions in terms of a user study on explanation necessity, and additional annotations to the existing BDD-A data.

Weaknesses:
- Much of the paper is just common sense. Obviously, people are going to demand explanations when the scenarios are of the "near crash" type, and obviously, people do not need explanations for why a self-driving car is slowing down in front of a stop sign. The points made by the paper therefore seem trivial to me.
- The dataset assumption is a little strong, especially point 2: "The car action in the video is safe. In other words, the car action should not put the passengers into a high-risk car accident." On the contrary, dangerous self-driving actions are really the ones that need explanations. For example, while people do not need explanations for why a self-driving car is slowing down in front of a stop sign, people DO need explanations for why a self-driving car does NOT slow down in front of a stop sign.
- How will the dataset be used to improve human-autonomous vehicle interactions? Please explain and give an example.

---

### Official Review · Reviewer_6FPf · 2022-10-16
**Good addition to the TEA workshop!**

**Rating:** 8
**Confidence:** 4

**Review:**

Summary
---
This paper addresses the question of when an AV should yield an explanation to a human driver. It looks at which driving scenarios necessitate explanations, and how this varies across drivers.

Strengths
---
This is a very compelling question to study!
The contribution of a dataset and the critique of the DeepDrive eXplanation Dataset are great.

Proposed Improvements
---
An implicit assumption in this work is that explanations are for AV passengers. But explanations may be necessary for other stakeholders - such as the US’s NHTSA. It would be good to clarify this point in the text.
I’d like to know more about the clustering approach for extracting diverse and representative scenarios from the dataset! Equally, there is some literature on Quality Diversity which might be helpful for this in the future, e.g., https://arxiv.org/abs/2012.04283

Other comments
---
Increased trust and acceptance is not necessarily a benefit! We want *calibrated* trust and acceptance!

I find it surprising that aggressive drivers don’t desire explanations, especially in light of the paper, “Do You Want Your Autonomous Car To Drive Like You?” by Basu et al. I'd recommend looking at this paper and seeing where you find differences, and questioning why you see those differences.

---

### Official Review · Reviewer_WGcB · 2022-10-17
**Good insights on the need for explanations for AV decisions**

**Rating:** 8
**Confidence:** 4

**Review:**

This paper presents findings on the need for explanations of AV decisions. The key insight is that passengers do not necessarily desire an explanation for every scenario. Explanations are strongly desired only in near crash or lane merging scenarios or those where the car behaves differently than the passengers’ expectation.

These insights are essential for AVs, both from a development standpoint and also user experience. This paper presents a good initial step towards this direction -- I recommend its acceptance.

---

### Decision · Program_Chairs · 2022-10-21

**Decision:**

Accept

**Comment:**

The paper is well-written and focus on a very important and interesting question for autonomous driving. All the reviewers have positive attitudes towards the paper. Please consider the comments from the reviewers in the final version.